

# Systematic analysis of NAC transcription factors in *Gossypium barbadense* uncovers their roles in response to Verticillium wilt

Zhanji Liu, Mingchuan Fu, Hao Li, Yizhen Chen, Liguo Wang and Renzhong Liu

Key Laboratory of Cotton Breeding and Cultivation in Huang-Huai-Hai Plain, Ministry of Agriculture, Cotton Research Center of Shandong Academy of Agricultural Sciences, Jinan, China

Corresponding author
Zhanji Liu, scrcliuzhanji@sina.com

## ABSTRACT

As one of the largest plant-specific gene families, the NAC transcription factor gene family plays important roles in various plant physiological processes that are related to plant development, hormone signaling, and biotic and abiotic stresses. However, systematic investigation of the *NAC* gene family in sea-island cotton (*Gossypium babardense* L.) has not been reported, to date. The recent release of the complete genome sequence of sea-island cotton allowed us to perform systematic analyses of *G. babardense NAC GbNAC*) genes. In this study, we performed a genome-wide survey and identified 270 *GbNAC* genes in the sea-island cotton genome. Genome mapping analysis showed that *GbNAC* genes were unevenly distributed on 26 chromosomes. Through phylogenetic analyses of GbNACs along with their Arabidopsis counterparts, these proteins were divided into 10 groups (I–X), and each contained a different number of GbNACs with a similar gene structure and conserved motifs. One hundred and fifty-four duplicated gene pairs were identified, and almost all of them exhibited strong purifying selection during evolution. In addition, various *cis*-acting regulatory elements in *GbNAC* genes were found to be related to major hormones, defense and stress responses. Notably, transcriptome data analyses unveiled the expression profiles of 62 *GbNAC* genes under Verticillium wilt (VW) stress. Furthermore, the expression profiles of 15 *GbNAC* genes tested by quantitative real-time PCR (qPCR) demonstrated that they were sensitive to methyl jasmonate (MeJA) and salicylic acid (SA) treatments and that they could be involved in pathogen-related hormone regulation. Taken together, the genome-wide identification and expression profiling pave new avenues for systematic functional analysis of *GbNAC* candidates, which may be useful for improving cotton defense against VW.

## INTRODUCTION

The plant-specific *NAC* genes (U̲N̲AM, no apical meristem; A̲TAF, Arabidopsis transcription activation factor; and C̲UC, cup-shaped cotyledon) form one of the largest families of transcription factors (*Nuruzzaman, Sharoni & Kikuchi, 2013*). Typically, NAC proteins

harbor a highly conserved NAC domain at the N-terminal and a variable transcriptional regulatory region (TR) at the C-terminal. The NAC domain can be further divided into five subdomains (A–E) and functions as DNA binding, nuclear localization, and formation of homodimers or heterodimers, while the TR region is responsible for transcription regulation as either an activator or a repressor (*Olsen et al., 2005*). *NAM*, the first *NAC* gene, was discovered in Petunia and functions in determining positions of shoot apical meristems and primordia (*Souer et al., 1996*). Since then, a large number of *NAC* genes have been identified from diverse plant species (*Nuruzzaman et al., 2010*; *Liu et al., 2019*).

Accumulated evidences indicate that the functions of *NAC* genes are associated with almost every biological process in plants, such as leaf senescence (*Fan et al., 2015*; *Zhao et al., 2016*), secondary cell wall formation (*Zhang et al., 2018*), and hormone signaling (*Takasaki et al., 2015*). Notably, a number of NAC genes, especially ATAF subfamily members, were proven to serve as critical regulators in plant defense against biotic and abiotic stresses (*Nuruzzaman, Sharoni & Kikuchi, 2013*; *Karanja et al., 2017*). In Arabidopsis, *ANAC032* was induced by bacterial pathogen *Pst* (*Pseudomonas syringae* pv. *tomato* DC3000) infection, and SA and jasmonic acid (JA) treatments. Furthermore, transgenic Arabidopsis plants overexpressing *ANAC032* showed strongly enhanced resistance to *Pst*, while the *ANAC032* knockout mutant exhibited increased susceptibility to *Pst* (*Allu et al., 2016*). In rice, the expressions of *SNAC1* were enhanced under drought, salt, and cold stresses. Overexpression *SNAC1* in rice resulted in increased tolerance to drought (*Hu et al., 2006*). Similarly, transgenic *OsNAC111* showed improved resistance to blast fungus in rice by regulating the expression of several defense-related genes (*Yokotani et al., 2014*).

Cotton is the most important natural fiber crop, and is also used as a food crop due to high levels of vegetable oil and protein in cottonseeds (*Li et al., 2009*). However, cotton production can be dramatically decreased by the occurrence of VW, a devastating vascular disease caused by *Verticillium dahliae* (*Sun et al., 2013*). Typically, infected plants show leaf chlorosis, leaf shedding, vascular discoloration, wilting, and plant death. In general, upland cotton (*Gossypium hirsutum*), accounting for about 90% of annual world cotton production, is susceptible to VW, whereas sea-island cotton, accounting for approximately 5% of annual world cotton output, is immune to VW. Thus, extensive efforts have been made to investigate the molecular mechanism of sea-island cotton resistance to VW (*Zhang et al., 2015*). Recently, a *NAC* gene *GbNAC1* was identified from *G. barbadense*. Overexpression of *GbNAC1* in Arabidopsis can significantly enhance resistance to VW, implying that *G. barbadense NAC* genes might play pivotal roles in biotic stress resistance (*Wang et al., 2016*). The availability of diploid and tetraploid cotton genomes has allowed scientists to identify the *NAC* gene family members at the genome-wide scale, such as 145 genes in *G. raimondii* (*Shang et al., 2013*), 141 in *G. arboreum* (*Shang et al., 2016*; *Fan et al., 2018*), and 283 in *G. hirsutum* (*Sun et al., 2018*). However, systematic analysis of *NAC* genes in *G. babardense* has not been completed. *G. babardense* (AD$_2$) and *G. hirsutum* (AD$_1$) are allotetraploids, which evolve from transoceanic hybridization between an A-genome species immigrated from Africa and a native American D-genome species, while *G. arboreum* (A$_2$) and *G. raimondii* (D$_5$) are diploids resembling the A-genome

and D-genome progenitor, respectively (*Hu et al., 2019*). In this study, we performed a genome-wide survey and identified 270 *GbNAC* genes in the latest *G. babardense* genome released by *Wang et al. (2019)*. These *GbNAC* genes were classified into 10 groups on the basis of sequence similarity. Sequence comparison among *GbNAC* genes revealed the presence and distribution of duplicated genes. Additionally, to identify *GbNAC* candidate genes associated with VW resistance, we analyzed the expression patterns of *GbNAC* genes using available transcriptome data of *G. babardense* cv. 7,124 inoculated with the fungal pathogen *V. dahliae*. Subsequently, qPCR-based gene expression profiling demonstrated that selected *GbNAC* candidate genes might be involved in MeJA and SA regulation. This study provides comprehensive information about sea-island cotton *NAC* genes, as well as a foundation for in-depth functional analysis of novel *GbNAC* candidate genes, which may be useful for the improvement of pathogen resistance in cotton.

## MATERIALS AND METHODS

### Identification of *NAC* genes in the sea-island cotton genome

We downloaded the genome sequences (v. HAU) of sea-island cotton from the CottonGen database (https://www.cottongen.org/, *Wang et al., 2019*). The Hidden Markov Model (HMM) profile of the NAC domain (PF02365) was retrieved from the Pfam database (http://pfam.xfam.org/, *Finn et al., 2016*). The HMMER program was used to search NAC protein in the sea-island cotton genome (*Finn, Clements & Eddy, 2011*). with an $E$-value cutoff of $e^{-5}$. Then, all the putative proteins were confirmed by the Pfam and SMART database (http://smart.embl-heidelberg.de/, *Letunic & Bork, 2018*). The MW (molecular weight) and pI (theoretical isoelectric point) of each NAC protein were predicted by the online software ExPASy (https://www.expasy.org/, *Artimo et al., 2012*). The TMHHM server (v. 2.0, http://www.cbs.dtu.dk/services/TMHMM/) was used to identify membrane-bound NAC proteins (*Krogh et al., 2001*). CELLO (v. 2.5, subCELlular Localization predictor, http://cello.life.nctu.edu.tw/; *Yu et al., 2006*) was used to predict the subcellular localization of GbNACs.

### Multiple alignments, phylogenetic analysis, gene duplication and synteny analysis

Multiple sequence alignments were performed with the NAC domain sequences of the GbNAC proteins using MEGA X (https://www.megasoftware.net/, *Kumar et al., 2018*). A phylogenetic tree was constructed by the neighbor-joining method with the following parameters: Poisson correction, pairwise deletion, and 1,000 bootstrap replicates. Gene duplications were analyzed with two major criteria, that is, the length of the aligned sequence covers more than 75% of the longer gene and similarity of the aligned regions is greater than 75% (*Vatansever et al., 2016*). Alignment of the coding sequences of duplicated genes was performed by the Clustal X (v. 2.0) program (*Larkin et al., 2007*), and the values of nonsynonymous (Ka) and synonymous (Ks) substitution rates were calculated using KaKs_Calculator package (*Zhang et al., 2006*) *via* model averaging. The approximate date of duplication events (million years ago, Mya) was estimated using the formula $T = Ks/2\lambda \times 10^{-6}$, on the basis of molecular clock rate of $2.6 \times 10^{-9}$ substitutions/synonymous site

for cotton (*Liu et al., 2015*). The relationships of duplicated genes were illustrated with the Circos program (*Krzywinski et al., 2009*). MCScanX was used to detect the synteny of *NAC* genes between *G. barbadense* and the other plant species (*Wang et al., 2012*).

## Gene structure, chromosomal mapping, and conserved motif analysis

The gene structures were determined using the CDS and DNA sequences of *GbNAC* genes and visualized by the Gene Structure Display Server (http://gsds.cbi.pku.edu.cn/, *Hu et al., 2015*). The positions of these *GbNAC* genes were determined by using the nucleotide sequence as a query to search against the *G. barbadense* genome. In addition, chromosomal localization map was constructed by using the MapChart (v. 2.32) program (*Voorrips, 2002*). In order to identify the conserved motifs among all the *GbNAC* genes, their protein sequences were subjected to the online software MEME (http://meme-suite.org/tools/meme, *Bailey et al., 2015*) using default parameters with exception for number of motif. The number of motifs was set to 20.

## *Cis*-acting regulatory element and miRNA target analysis

For *cis*-acting regulatory element analysis, we retrieved 1,500 bp DNA sequences up-stream from the transcription start site from the newly released *G. barbadense* genome sequences (*Wang et al., 2019*) and then screened them in the PlantCare (http://bioinformatics.psb.ugent.be/webtools/plantcare/html/, *Lescot et al., 2002*) and PLACE (https://www.dna.affrc.go.jp/PLACE/, *Higo et al., 1999*) databases. For miRNA target analysis, we downloaded all the mature miRNA sequences of Gossypium from the miRBase database (v. 22.1) (http://www.mirbase.org/). The online sever psRNATarget (http://plantgrn.noble.org/psRNATarget/, *Dai, Zhuang & Zhao, 2018*) was used to predict the miRNA target.

## Gene expression analysis under Verticillium wilt stress

The transcriptome data of sea-island cotton were obtained from the Sequence Read Archive under accession number SRP03537 at the NCBI website (*Chen et al., 2015*), where the plants were inoculated with *V. dahliae*. Briefly, two-week-old seedlings of *G. barbadense* resistant cultivar 7,124 were inoculated with the high virulence V991 defoliating strain of *V. dahliae* ($5 \times 10^6$ spores/mL) by the root-dip method for 2, 6, 12, 24, 48, and 72 h. Then, the samples, including the mock-inoculated (control) and six inoculated, were collected for RNA sequencing. We retrieved the expression data of *GbNAC* genes from the root under *V. dahliae* infection (*Chen et al., 2015*). The hierarchical clustering and the heatmap-based expression profiles of *GbNAC* genes were performed using ClustVis (*Metsalu & Vilo, 2015*). The Venn map of differentially expressed *GbNAC* genes was constructed using the UpSetR package (*Lex et al., 2014*).

## RNA isolation and qPCR analysis

*G. barbadense* cv. 7,124 seeds were cultivated in commercial soil at 28 °C with a photoperiod of 16 h light/8 h dark. Two-week-old seedlings were gently uprooted, rinsed and cultivated in Hoagland solution for two days. Then these seedlings were treated with Hoagland

solution containing 0.1 mM MeJA and 1 mM SA, respectively. Roots were sampled from three biological replicates after treatment for 1, 2, 6, and 12 h, then immediately immersed in liquid nitrogen and stored at −80 °C for qPCR.

The total RNA from the root samples treated with MeJA and SA and the control (roots from Hoagland solution without hormone) was extracted using Trizol (Invitrogen). The first strand cDNA was generated from 1 μg of total RNA using a PrimerScript$^{TM}$ 1st Strand cDNA Synthesis Kit (Takara, Dalian, China). qPCR was performed with three replicates using an ABI QuantStudio 5 Real-Time PCR System (Thermo Fisher Scientific, Waltham, MA, USA) and SYBR Premix Ex Taq$^{TM}$ (Takara, Dalian, China). The amplification procedure was as follows: one cycle at 95 °C for 3 min; then 40 cycles at 95 °C for 15 s, 60 °C for 15 s. The cotton *actin* (AF059484) was selected as the internal reference gene (*Zhang et al., 2013*). Gene expression levels were calculated according to the $2^{-\Delta\Delta CT}$ method described by *Livak & Schmittgen (2001)*. The primers used for qPCR are listed in Table S1.

## RESULTS

### Identification and phylogenetic analysis of NAC family members in *Gossypium barbadense*

A total of 270 NAC proteins were identified from *G. barbadense* and named as GbNAC001 to GbNAC270. All of the 270 GbNACs contained the NAC domain (PF02365) based on Pfam and SMART tests. The lengths of GbNAC proteins ranged from 154 (GbNAC059) to 959 (GbNAC175) amino acids with MW from 17.65 to 107.75 kDa, and pI from 4.67 to 9.79. Subcellular localization of GbNACs was predicted using the online software CELLO (http://cello.life.nctu.edu.tw/). Among the 270 GbNAC proteins, three were predicted to be mitochondrial proteins (GbNAC031, 032, and 255); five were located in the chloroplast (GbNAC014, 036, 116, 174, and 249); 10 were extracellular; 23 were cytoplasmic, and the rest were localized in the nucleus. These results are similar to those of cucumber (*Liu et al., 2018*). Detailed information including gene locus, chromosome location, exon number, sequence length, MW, pI, Arabidopsis orthologous locus and subcellular location of all identified GbNAC proteins is provided in Table S2.

The NAC domain sequences of GbNAC proteins were used to construct a neighbor-joining phylogenetic tree. As a result, the 270 GbNACs were classified into 10 groups which were named as I to X (Fig. 1). Group VI contained the most NAC members with 47 GbNACs, followed by group I with 39 GbNACs. Group III had the least NAC members with only ten GbNACs. Additionally, 35 GbNACs with high similarity to Arabidopsis ATAF members were clustered in group IV. Arabidopsis ATAF members play pivotal roles in the responses to biotic and abiotic stresses. These results suggest that NAC members in group IV may have similar functions with Arabidopsis ATAF members.
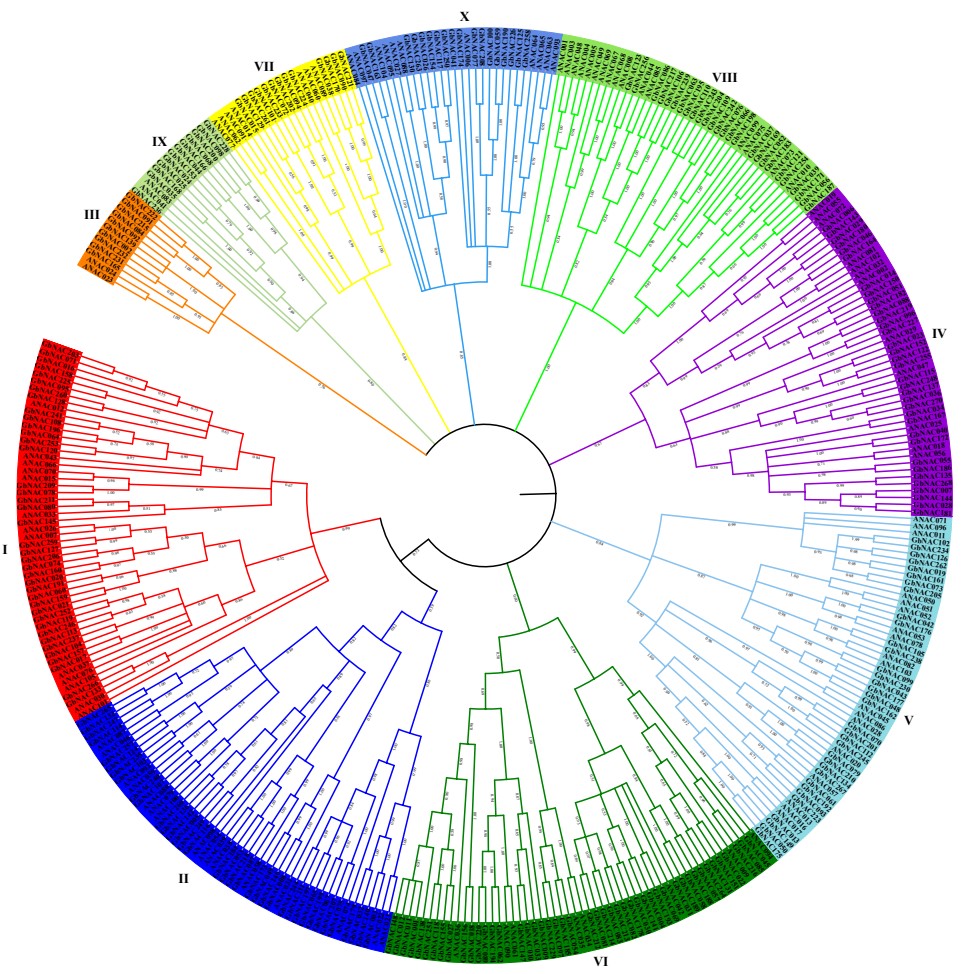

**Figure 1 Phylogenetic tree of the 270 GbNAC proteins.** Multiple sequence alignment of NAC domain sequences of *G. barbadense* and *Arabidopsis* was performed using ClustalW. MEGA X was used to construct the neighbor-joining (NJ) tree with 1000 bootstrap replicates. Various colors indicate different groups of GbNACs.

## Chromosomal locations, duplications, and synteny analysis of the *GbNAC* genes

To determine the chromosomal distribution of the *GbNAC* genes, we searched the sea-island cotton genome database using blastn and the DNA sequence of each *GbNAC* gene. The results suggested that 263 *GbNAC* genes (97.4%) were mapped to 26 chromosomes. Specifically, 132 *GbNAC* genes were distributed in the A-subgenome, and 131 were located in the D-subgenome (Fig. 2). In addition, five genes (*GbNAC011*, *GbNAC012*, *GbNAC049*, *GbNAC050*, and *GbNAC069*) were anchored in four A-subgenome scaffolds, and two genes (*GbNAC148* and *GbNAC262*) were found in two D-subgenome scaffolds. The number of *GbNAC* genes distributed on each chromosome was uneven. Chromosome D11 contained the highest number of *NAC* genes, with 15 *GbNAC* genes. In contrast, chromosome A10 contained the least number of *NAC* genes, with only five genes. Interestingly, many *GbNAC*

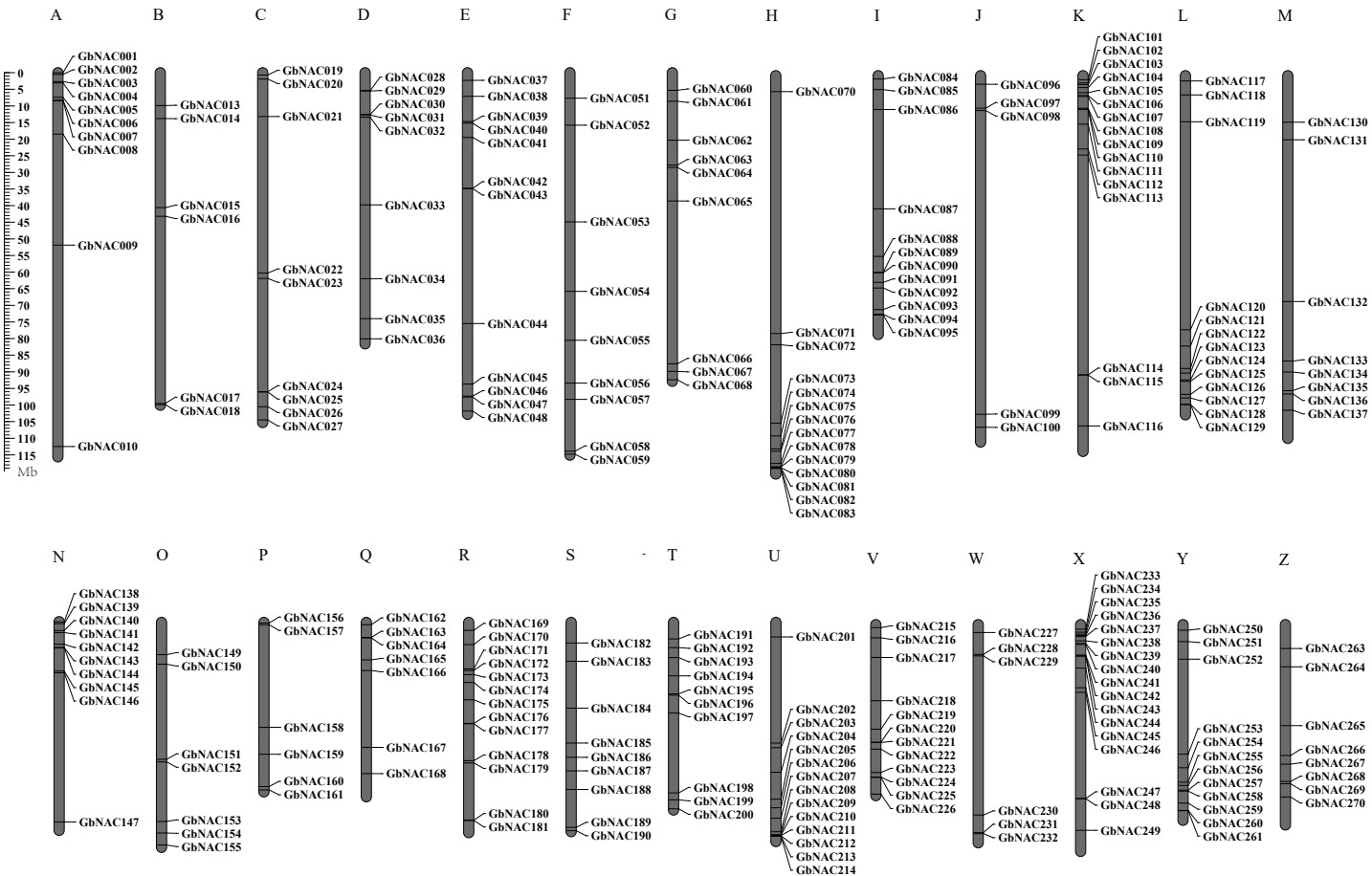

**Figure 2   Chromosomal locations of the *GbNAC* genes.** Grey bars denote the *G. babardense* chromosome. Scale bar on the left indicates the chromosome lengths (Mb). The letter A-M indicates the A-subgenome chromosome A01–A13, and N-O indicates the D-subgenome chromosome D01–D13, respectively.

genes were clustered within a short distance, such as the top of A11 and D11 and the bottom of A08 and D08.

Gene duplication including tandem duplication, segmental duplication, and whole-genome duplication (WGD) is a major driving force in the evolution of plants. The origin of multigene family is due to gene duplication that arose from region-specific duplication or WGD (*Du et al., 2012*). To reveal the expansion mechanism of the *GbNAC* gene family, gene duplication analysis was performed using blastn and the coding sequences (cds) of all *GbNAC* genes. In all, we identified 148 pairs (212 *GbNAC* genes) of segmental duplications, three pairs of tandem duplications (GbNAC011/GbNAC012, GbNAC024/GbNAC025, and GbNAC082/GbNAC083), and one triplicate repeat of tandem duplications (GbNAC212/GbNAC213/GbNAC214) (Fig. 3). One hundred and thirty-two duplication gene pairs occur between the A-subgenome and the D-subgenome, while only 12 and 10 duplication gene pairs occur within the A-subgenome and the D-subgenome, respectively. These genes represent approximately 81.9% (221 of 270) of the *GbNAC* genes,
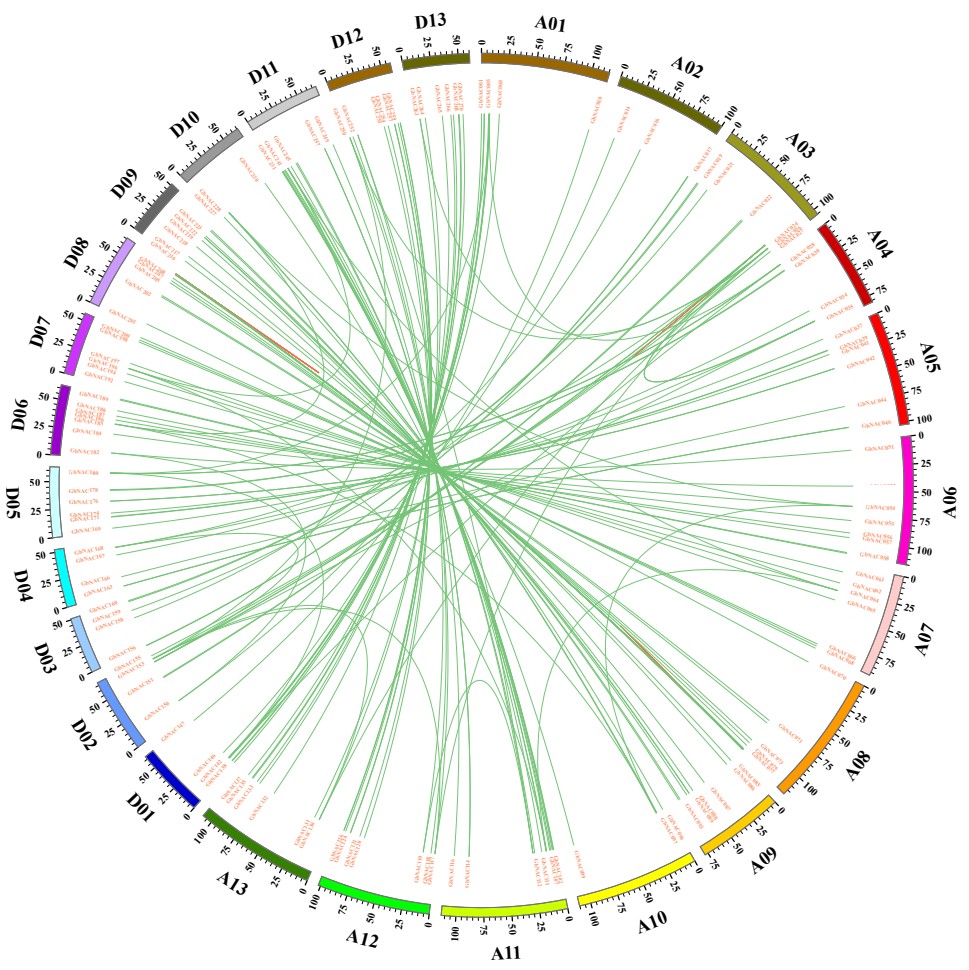

**Figure 3  Circos diagram of the *GbNAC* duplication pairs in *G. barbadense*.** 143 *GbNAC* duplication pairs are linked with green lines. Scale bar marked on the chromosome indicating chromosome lengths (Mb).

indicating their origin may be from sea-island cotton genome duplication events. Using the Ka and Ks of each duplicated *GbNAC* gene pair, we found that the Ks values of all gene pairs were between 0.008 and 0.912. Specifically, the Ks values of 87 (58.78%) gene pairs were less than 0.05. Additionally, the Ka/Ks value of each gene pair was calculated and the Ka/Ks values of 140 gene pairs (94.59%) were less than 1, which indicated these genes had evolved under strong purifying selection. Furthermore, eight gene pairs (Ka/Ks >1) may evolve under strong positive selection after duplication. Moreover, we also calculated the approximate date of duplication events. The duplication events of *GbNAC* genes occurred from 1.58 Mya (Ks = 0.008) to 175.29 Mya (Ks = 0.912), with a mean of 59.31 Mya (Ks = 0.154). Detailed information including duplication gene pairs, chromosome location, duplication type, Ka, Ks, Ka/Ks and approximate duplication date (Mya) of all identified duplicated *GbNAC* genes is provided in Table S3.

To detect the synteny of *NAC* genes, we performed a collinearity analysis between *G. barbadense* and the other four plant species (*Arabidopsis thaliana*, *G. arboretum*, *G. raimondii* and *G. hirsutum*) using MCScanX. Previously, 117, 141, and 145 *NAC* genes were revealed in *A. thaliana* (AtNAC, Nuruzzaman et al., 2010), *G. arboretum* (GaNAC, Shang et al., 2016), and *G. raimondii* (GrNAC, Shang et al., 2013), respectively. For *NAC* genes in *G. hirsutum* (GhNAC), we identified 272 *GhNAC* genes in the upland cotton genome that was released recently (Wang et al., 2019). Totally, 945 *NAC* genes were used to evaluate synteny relationship in this study. As a result, we found 421 paired collinearity relationships between 247 *GbNAC* and 234 *GhNAC* genes, 241 pairs between 241 *GbNAC* and 133 *GrNAC* genes, 181 pairs between 120 *GbNAC* and 54 *AtNAC* genes, and 142 pairs between 141 *GbNAC* and 81 *GaNAC* genes (Fig. 4 and Table S4). Notably, 69 *GbNAC* genes are collinear with NAC genes from the other four species (Table S4).

### *GbNAC* gene structures and conserved motifs

To better understand the relationship between gene function and evolution among the *GbNAC* genes, the exon/intron organization and conserved motifs were analyzed (Fig. 5). The number of exons ranged from 1 to 10. Most *GbNAC* genes (182/270, 67.41%) had three exons (Fig. 5), although *GhNAC059*, *GhNAC105*, and *GhNAC238* contained only one exon, and *GhNAC129* and *GhNAC175* contained 10 exons, the highest number of all genes. We also found that *GbNAC* genes in the same group had a similar gene structure. For example, all 12 members in group IX had three exons. Among the 39 members in group I, all had three exons except *GbNAC060* and *GbNAC145* (Fig. 5).

Twenty conserved motifs were identified among the 270 GbNAC proteins (Fig. S1 and Table S5). As a result, all GbNAC proteins contain a conserved NAC domain at the N-terminal, which includes five subdomains (A–E, Fig. 6). Notably, the members with high similarity in the same group shared a common motif composition. For example, GbNAC213 and GbNAC214 were found to contain the same three motifs (Fig. 5). This finding indicates that these genes may have similar functions. Most GbNAC proteins (257/270, 95.19%) contain 2–5 conserved motifs. However, seven GbNAC proteins (GbNAC033, GbNAC068, GbNAC092, GbNAC109, GbNAC200, GbNAC226, and GbNAC242) contain only one motif and six GbNAC proteins (GbNAC017, GbNAC020, GbNAC074, GbNAC160, GbNAC206, and GbNAC253) contain six motifs. Additionally, we found that subdomains A (246/270), and/or C (200/270), and/or D (250/270) are present in most GbNAC proteins. Furthermore, we found that 231 GbNAC proteins contained the conserved histidine residue in subdomain D, which influences homodimerization and DNA binding of NAC proteins (Kang et al., 2018).

Membrane-bound NAC proteins feature a distinctive transmembrane motif (TM) at either the C terminal region or the N terminal region and play vital roles in plant defense against abiotic stresses (Kim et al., 2010; Li et al., 2016; Sun et al., 2018). In the sea-island cotton genome, 22 membrane-bound GbNAC proteins were identified (Table S2). Notably, 20 membrane-bound GbNAC proteins contain one TM at the C terminal and two GbNAC members (GbNAC110 and GbNAC243) contain one TM at the N terminal.

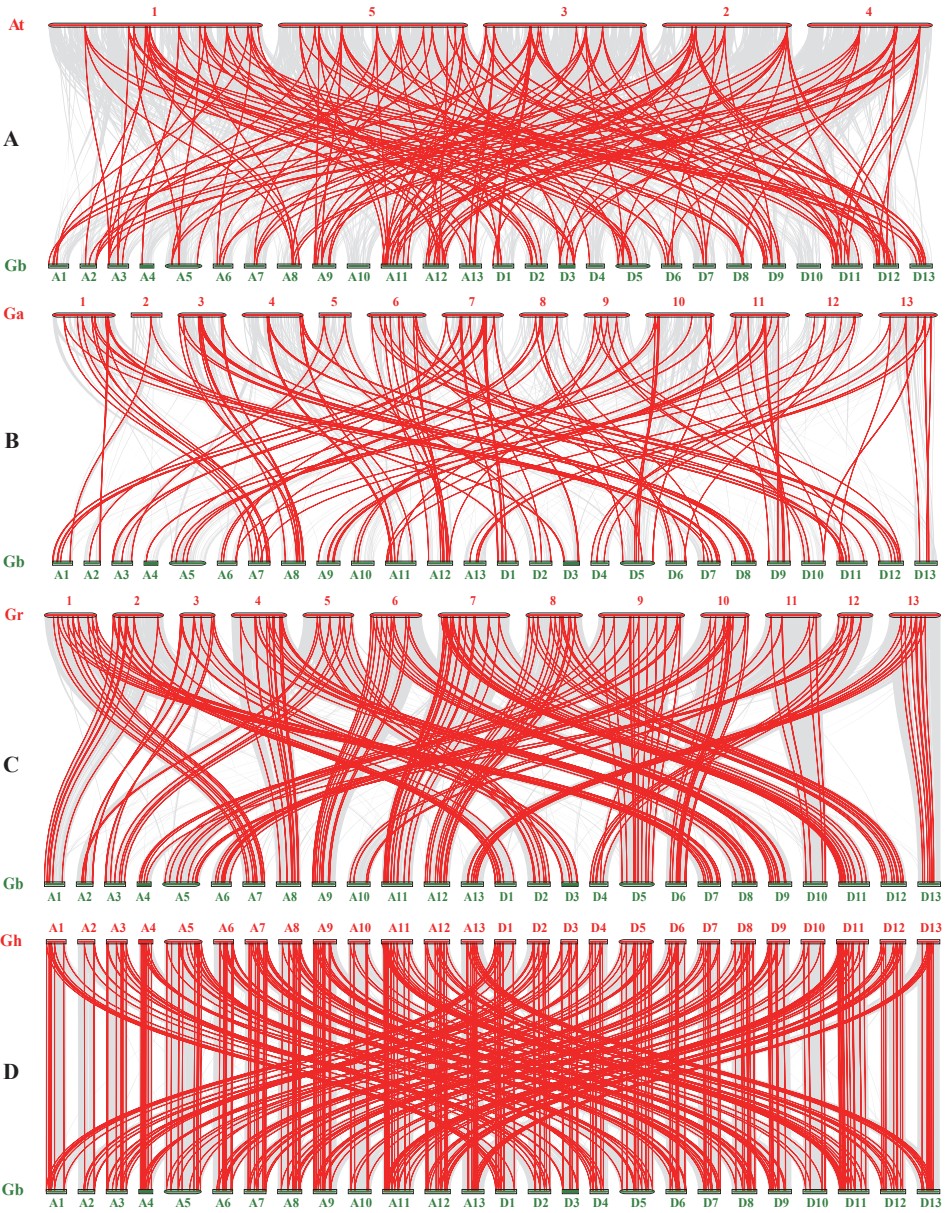

**Figure 4   Synteny analysis of NAC genes between Gossypium barbadense and other plant species.** At, Ga, Gr, Gb, and Gh indicate *Arabidopsis thaliana*, *G. arboreum*, *G. raimondii*, *G. barbadense*, and *G. hirsutum*, respectively. (A) Collinearity between At and Gb; (B) Collinearity between Ga and Gb; (C) Collinearity between Gr and Gb; (D) Collinearity between Gh and Gb. Gray lines in the background represent the collinear blocks within the genomes of *G. barbadense* and other plant species, while the red lines show the collinear *NAC* gene pairs.

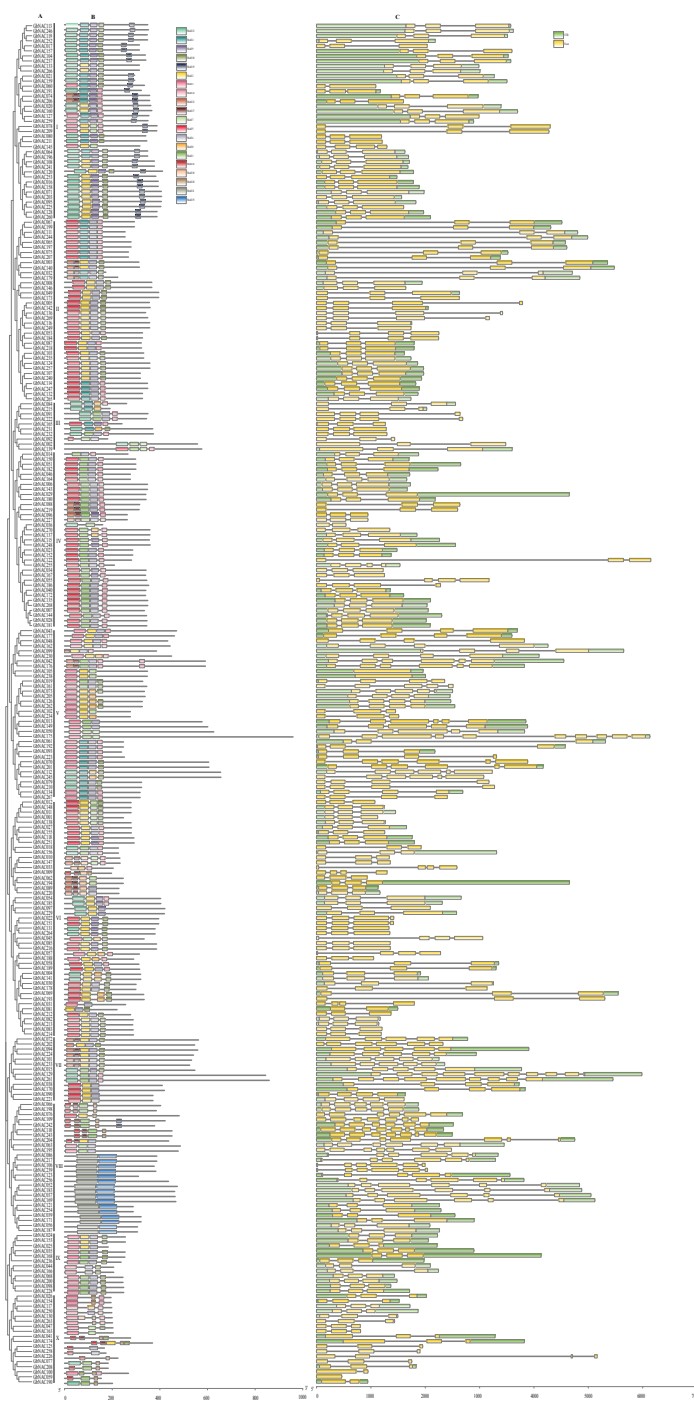

**Figure 5  Putative conserved motifs and gene structures of the *GbNAC* genes.** (A) Phylogenetic tree. Multiple sequence alignment of NAC domain sequences of *G. barbadense* was performed using ClustalW. The neighbor-joining (NJ) tree was constructed using MEGA X with 1,000 bootstrap replicates. (B) Conserved motif. MEME analysis revealed the conserved motifs of the GbNAC proteins. The colored boxes on the right denote 20 motifs. (C) Gene structure. The yellow boxes, black lines, and green boxes represent exon, intron, and UTR (untranslated region), respectively.

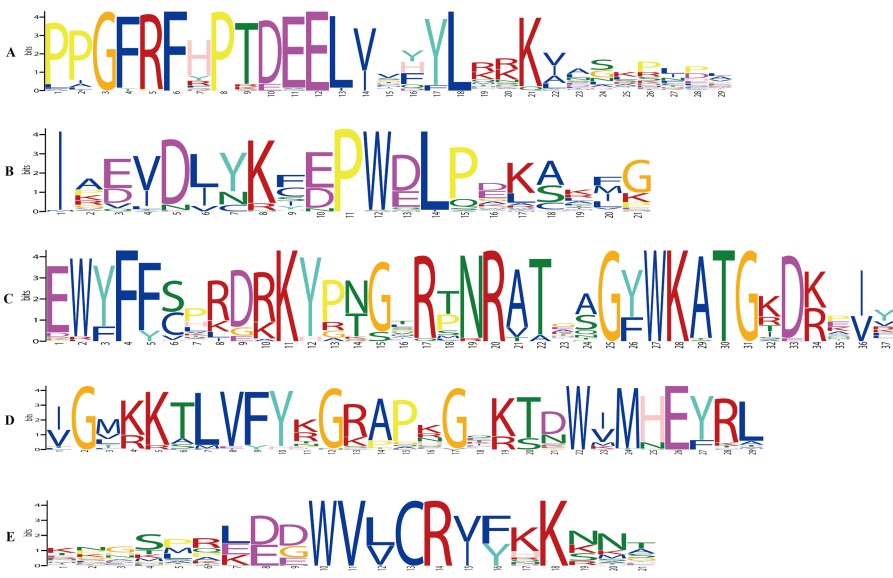

**Figure 6   Conserved subdomains in the NAC domain at the N-terminal of GbNAC proteins.** A–E indicate five conserved subdomain A–E, respectively.

These membrane-bound GbNAC proteins are distributed in three groups: seven in group V, 11 in group VII, and four in group VIII.

## Cis-acting regulatory element and miRNA target analysis

To better understand the transcriptional regulation mechanisms of *GbNAC* genes, we characterized the *cis*-acting regulatory elements within a 1,500 bp upstream region from the transcription start site using PlantCARE and PLACE database (Table S6). A large number of *cis*-acting regulatory elements were identified in promoter sequences of 270 *GbNAC* genes (Fig. S3 and Table S7). Common regulatory elements such as TATA-box and CAAT-box were present in all *GbNAC* genes. Meanwhile, we identified 11 *cis*-acting regulatory elements related to hormone responses. Among these, AuxRR-core and TGA-element, auxin-responsive element; ABRE, MYB, and MYC, *cis*-element involved in abscisic acid (ABA) responsiveness; GARE, P-box, and TATC-box, involved in gibberellin responsiveness; ERE, ethylene responsive element; CGTCA-motif, MeJA responsive elements; and TCA-element, involved in SA responsiveness. Notably, all of the 270 *GbNAC* genes contained at least one hormone-responsive element (Fig. S3). MYB, MYC, and ERE were available in at least 80% of the *GbNAC* genes. Additionally, promoter sequences of some *GbNAC* genes also contained several elements involved in biotic and abiotic stress responses, including pathogen defense (AT-rich and TC-rich repeat), drought (MBS), cold (DRE and LTR), anaerobic stress (ARE), and wounding (WUN-motif). Thus, *GbNAC* genes could be regulated by diverse hormone and environmental changes.

Recent reports have defined a subset of genes from the NAC-domain gene family as potential targets of miRNAs. To determine the involvement of miRNAs in regulating the expression of *GbNAC* genes, putative miRNA targets were determined in the 270

*GbNAC* genes using the online sever psRNATarget with the default parameters except Expection (≤3.0). Totally, 43 *GbNAC* genes were predicted as the targets of 21 known miRNAs (Table S8). Seven *GbNAC* genes were each predicted to be the targets of two miRNAs and *GbNAC007* was the target of three miRNAs (gra-miR8634, gra-miR8786a and gra-miR8786b). Specifically, miR164 targets 14 *GbNAC* genes, which are all from group II (Table S8). In Arabidopsis, the miR164 family (ath-miR164a/b/c) guides the cleavage of the transcripts of five *NAC* genes (*NAC1/At1G56010*, *CUC1/At3g15170*, *CUC2/At5g53950*, *ANAC080/At5g07680*, and *ANAC100/At5g61430*) that function in the regulation of plant growth and development such as lateral root emergence, formation of vegetative and floral organs, and age-dependent cell death (*Fang, Xie & Xiong, 2014*; *Hernandez & Sanan-Mishra, 2017*). The 14 miR164-targeted *GbNAC* genes and the five Arabidopsis *NAC* genes were clustered into group II, indicating that these genes have high sequence identity and may have a similar function.

## Expression profile of *GbNAC* genes in response to Verticillium wilt and hormones

We used publicly available transcriptome data to assess expression of *GbNAC* genes in roots under VW stress. As a result, 239 (88.52%) *GbNAC* genes were identified to be expressed in infected root samples (Fig. 7). Three patterns (Pattern I–III) of expression were revealed. Genes from Pattern I have low expression levels in the control and then were up-regulated gradually during VW infection. Genes from Pattern II, in contrast to Pattern I, were highly expressed in the control and then were subsequently down-regulated during VW infection. Genes from Pattern III showed high expression levels in the control, then were down-regulated at the early stages (2, 6 and 12 h) of inoculation, and finally up-regulated at the late stages (24, 48 and 72 h) of inoculation. There are 130 duplicated gene pairs among the 239 *GbNAC* genes. Most duplicated gene pairs (80.77%) demonstrated similar expression patterns, suggesting that duplicated genes are functionally redundant. However, some duplicated genes have a divergent expression. For example, *GbNAC082* and *GbNAC083* are tandem duplication genes. *GbNAC082* had considerably high expression in the control and was up-regulated by VW stress at 2 h, while *GbNAC083* had low expression in the control and was down-regulated remarkably by VW stress at 2 h.

The expression of 192 *GbNAC* genes was significantly altered ($|\log_2^{\text{Fold}}| \geq 1$) in roots inoculated with *V. dahliae* as compared to the control in at least one time point (Fig. S2). 103, 130, 123, 135, 130, and 128 *GbNAC* genes were significantly induced at 2, 6, 12, 24, 48, and 72 h after inoculation, respectively, which indicates the number of differentially expressed *GbNAC* genes was similar among different time points. Additionally, the expression of 62 *GbNAC* genes including 15 up-regulated and 47 down-regulated genes were significantly altered at all of the six time points (Fig. S2 and Table S9). Eighteen duplicated gene pairs were revealed among the 62 *GbNAC* genes. Notably, *GbNAC* genes in each duplicated gene pair had similar expression patterns. The expression of previously reported sea-island cotton *NAC* gene *GbNAC1* (*GbNAC220* in this study) was also altered. *GbNAC1*, a positive regulator involved in cotton resistance to *V. dahliae*, was down-regulated in infected roots compared to the control, which is consistent with qPCR

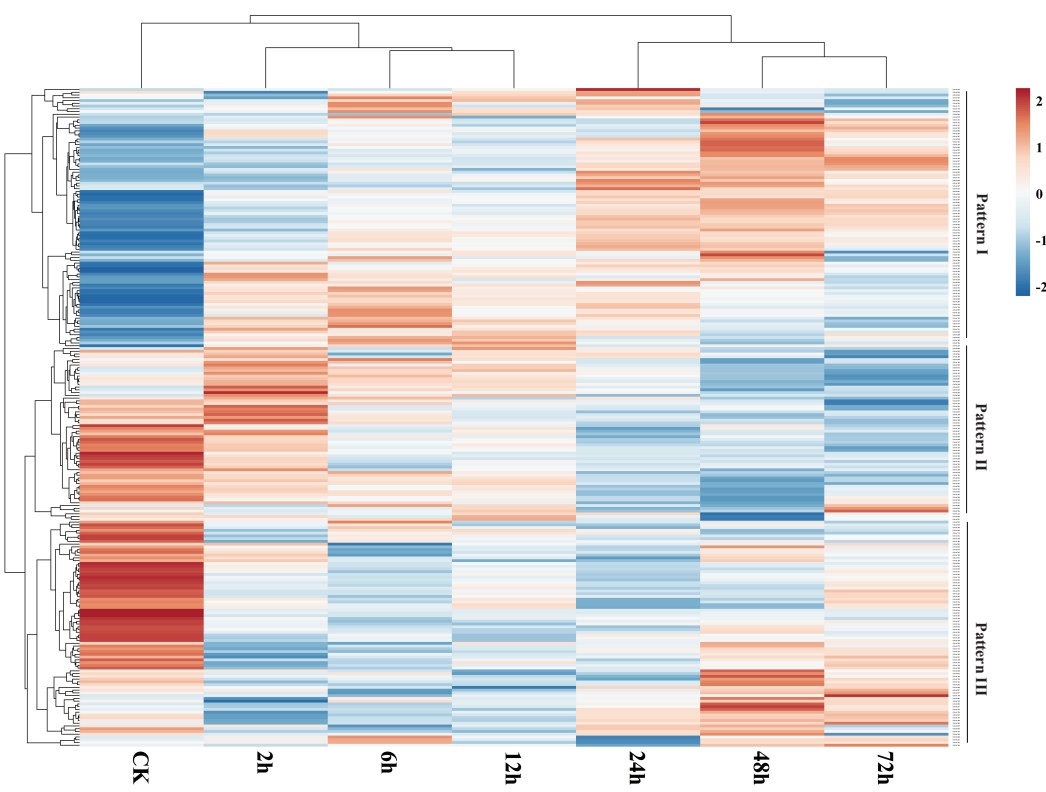

**Figure 7** **Expression profiles of *GbNAC* genes in roots under Verticillium wilt infection.** The heat map was generated by ClustVis software. The expression data were gene-wise normalized and hierarchically clustered with average linkage. The bar on the right of the heat map indicates relative expression values. Values 2, 0, −2 represent high, intermediated, and low expression, respectively.

results reported by *Wang et al. (2016)*. These results imply that *GbNAC* genes could play crucial roles in the defense of cotton against VW.

Phytohormones, such as MeJA and SA, regulate plant defenses against diverse pathogens. In order to identify hormone-responsive *GbNAC* genes, *G. barbadense* cv. 7124 seedlings were treated with MeJA and SA, and the changes in transcript abundance of 15 genes selected from the 62 differentially expressed *GbNAC* genes were analyzed by qPCR. As shown in Fig. 8, all the *GbNAC* genes tested were sensitive to the hormones MeJA and SA, but the levels of sensitivity were substantially different at the four time points. In general, most tested *GbNAC* genes were up-regulated under MeJA treatment, but down-regulated under SA treatment. Specifically, the expression of two *GbNAC* genes (*GbNAC014* and *GbNAC164*) from group IV were up-regulated at all the four time points, and had at least 186-fold and 225-fold increase at 6 h compared with control, respectively (Fig. 8).

# DISCUSSION

## Characterization of *GbNAC* genes

In this study, we performed a genome-wide analysis of the sea-island cotton *GbNAC* gene family to investigate their potential functions in response to VW. As a result,

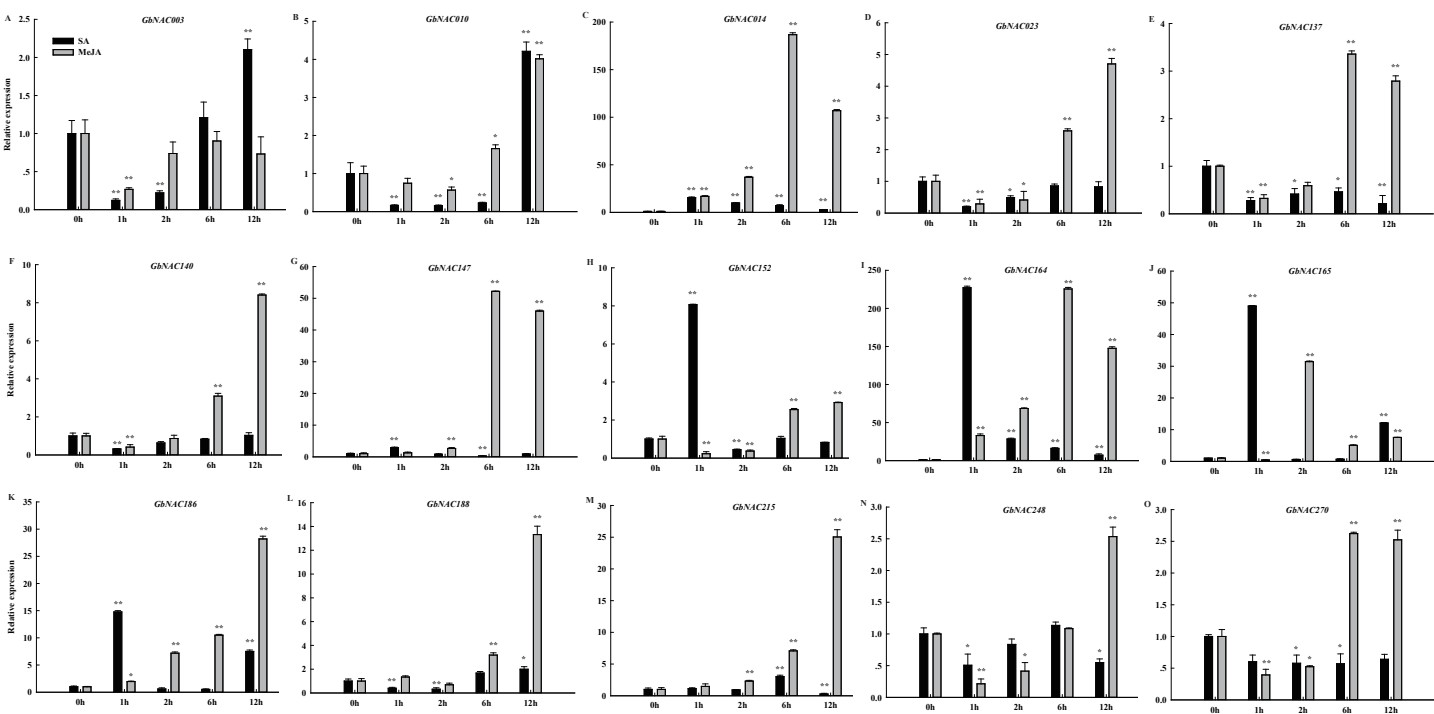

**Figure 8** Expression profiles of *GbNAC* genes in response to hormone MeJA and SA. qPCR was used to analyze the expression profiles of 15 *GbNAC* genes under hormone MeJA and SA treatment. A-O indicates the 15 *GbNAC* genes tested, respectively. The relative expression levels are normalized to *GbActin*. The data represent the mean of three biological replicates. Significant differences were indicated as * (Tukey's HSD, $P <$ 0.05) and ** ($P < 0.01$) above the bars between treatment and control.

270 *GbNAC* genes were revealed in the *G. barbadense* genome, which is similar to that found in *G. hirsutum* (283 *NAC* members, *Sun et al., 2018*), but is about twice as much as that found in *G. raimondii* (145, *Shang et al., 2013*) and *G. arboreum* (141, *Shang et al., 2016*; *Fan et al., 2018*). The difference in the size of *NAC* genes is because the genomes of tetraploid species possess more *NAC* genes than the diploid species, presumably because an allopolyploidization event occurred in *G. babardense* and *G. hirsutum* approximately 1.7–1.9 Mya (*Hu et al., 2019*). *GbNAC* genes were unevenly distributed on 26 chromosomes, and a number of *GbNAC* genes were clustered on the top or bottom of specific chromosomes (Fig. 2). These results are similar to those of the two diploid cotton species, *G. raimondii*, and *G. arboreum* (*Shang et al., 2013*; *Shang et al., 2016*).

According to the multiple sequence alignment, all the GbNAC proteins contain the highly conserved NAC domain with 150–160 amino acids, which can be further divided into five distinct subdomains (A–E) at the N-terminal. However, a number of GbNAC proteins have an atypical NAC domain pattern. For example, three GbNAC proteins (GbNAC068, GbNAC200, and GbNAC226) have only conserved subdomain A, while four GbNAC proteins (GbNAC033, GbNAC092, GbNAC109, and GbNAC242) have only subdomain D. In addition, the previously reported GbNAC1 lacks the conserved subdomains B, C, and E (*Wang et al., 2016*). NAC proteins lacking one to four subdomains were also observed
in Arabidopsis, rice, and radish (*Ooka et al., 2003*; *Nuruzzaman et al., 2010*; *Karanja et al., 2017*). Thus, the atypical NAC domain pattern appears to be common among NAC proteins from diverse plant species. Previous studies revealed that subdomains A, C, and D have higher levels of conservation than subdomains B and E, and play important roles in the function of NAC genes (Ooka et al., 2003). Subdomain A has the ability to form a helical structure and is involved in the formation of homodimers or heterodimers with other NAC domain proteins (*Jensen et al., 2010*). Subdomain D contains the nuclear localization signal and subdomain C is associated with DNA binding activity (*Hernandez & Sanan-Mishra, 2017*). In this study, most GbNAC proteins (196/270) contain subdomains A, C, and D, whereas only 48 GbNAC proteins have subdomain B (Fig. 5). Furthermore, we found 231 GbNAC proteins contained a conserved histidine residue in subdomain D. A recent study revealed that the conserved histidine residue is present in 80% of Arabidopsis NAC members and functioned as a switch to regulate both pH-dependent homodimerization and DNA binding (*Kang et al., 2018*). Thus, the subdomain D may function not only in nuclear localization but also in the formation of functional dimers and DNA-binding.

*NAC* gene duplication was investigated in sea-island cotton genome. One hundred and fifty-four duplicated *NAC* gene pairs including 148 segmental duplication pairs and 6 tandem duplication pairs were revealed in sea-island cotton (Fig. 3). Therefore, it can be concluded that segmental duplication dominates the expansion of the *NAC* gene family in the *G. barbadense* genome. Furthermore, most duplicated gene pairs had undergone strong purifying selection during evolution, indicating that purifying selection played pivotal roles in the confinement of the *GbNAC* gene functions, which was further confirmed by the similar expression pattern of most duplication gene pairs. Previous studies have showed that the A and D ancestor genomes diverged approximately 6.2–7.1 Mya (*Hu et al., 2019*). In this study, about half of the duplication events occurred after the divergence of the two diploid progenitors (Table S3). Synteny analysis indicated that the collinearity of *NAC* genes for the four *Gossypium* species is highly conserved. However, the level of collinearity is different. *G. barbadense* and *G. hirsutum* have the highest level of collinearity, while *G. barbadense* and *G. arboretum* have the lowest level of collinearity (Table S4). The difference is probably attributed to the genomic characteristics and evolution history of cotton. *G. barbadense* and *G. hirsutum* evolved from a common tetraploid ancestor and diverged approximately 0.4–0.6 Mya (*Hu et al., 2019*), which leads to high collinearity for the two tetraploid species. Nevertheless, the A subgenome of *G. barbadense* has large chromosome inversions in comparison with *G. arboretum* due to chromosomal rearrangement after allopolyploidization (*Wang et al., 2019*), which results in relatively low collinearity between *G. barbadense* and *G. arboretum*.

## Differential expression of *GbNAC* genes in response to Verticillium wilt

Cotton VW is a destructive soil-borne fungal disease and dramatically reduces the yield and quality of cotton. VW was first reported in Virginia, USA, in 1914 and now occurs worldwide. Over the past century, substantial efforts have been made to develop ways to control VW, and a number of genes, such as *Gh_A10G2076*, *GhATAF1*, *GbERF1*,

*GbWRKY1*, and *GbRLK*, have been revealed to be associated with VW resistance (*Li et al., 2017*; *He et al., 2016*; *Guo et al., 2016*; *Li et al., 2014*; *Jun et al., 2015*). However, whether the NAC genes could play possible role in response to VW in *G. barbadense* is still in question. In this study, 62 *GbNAC* genes were identified to be significantly up- or down-regulated in roots inoculated with *V. dahliae* at all six time-points (Fig. S2 and Table S9). These genes are all from the 10 phylogenetic groups, except group V, and contain at least one *cis*-regulatory element involved in stress responses (Table S7). We also found that *GbNAC* genes in each duplicated gene pair had similar expression patterns in response to VW infection. The functional redundancy of duplicated genes may be related to their similar gene structure, motifs, and *cis*-regulatory elements. In Arabidopsis, five *NAC* genes (*ANAC016*, *ANAC036*, *ANAC037*, *ANAC061*, *ANAC081*, and *ANAC091*) were significantly up-regulated after chitin treatment (*Libault et al., 2007*). Chitin is an elicitor of plant defense responses and its elicitation plays important role in plant defense to fungal pathogens. Among the 62 *GbNAC* genes, 10 genes clustered with *ANAC081* are from group IV; 13 genes along with *ANAC036* and *ANAC061* belong to group VI; 4 genes and *ANAC091* are from group VII, and 3 genes and *ANAC037* are from group I (Table S9).

Previously, an upland cotton ATAF subfamily *NAC* gene, *GhATAF1* (DT549350), was reported to be up-regulated by *V. dahliae* inoculation. Cotton plants overexpressing *GhATAF1* increased susceptibility to pathogen *V. dahliae* (*He et al., 2016*). In our study, *GbNAC164*, an ortholog to *GhATAF1*, was down-regulated by *V. dahliae* infection. The different expression pattern may be caused by cotton genotypes and/or *V. dahliae* strains. *GbNAC164* was investigated in *G. barbadense* cv. 7,124 treated with a highly virulent *V. dahliae* strain V991, while *GhATAF1* was analyzed in *G. hirsutum* cv. YZ1 treated with a moderately aggressive *V. dahliae* strain ICD3-2. Sea-island cotton *GbNAC1* (*GbNAC220* in this study) belongs to the Tobacco elicitor-responsive gene encoding NAC-domain protein (TERN) subgroup and was down-regulated by VW. Cotton plants silencing *GbNAC1* reduce resistance to VW, whereas transgenic Arabidopsis lines overexpressing *GbNAC1* enhance resistance to VW compared to wild type (*Wang et al., 2016*). *GbNAC220* was also down-regulated by VW in our study. In addition, MeJA and SA are important pathogen-related hormonal regulators. Among the 62 *GbNAC* genes, 31 and 22 genes contain MeJA- and SA-responsive elements, respectively, and 12 genes contain both MeJA- and SA-responsive elements (Table S7), indicating the expression of these genes may be regulated by MeJA and/or SA. These results were further verified by qPCR analysis. The qPCR results indicated *GbNAC003*, *GbNAC137*, *GbNAC140*, *GbNAC215*, and *GbNAC248* were sensitive to MeJA and SA treatments (Fig. 8), which was in agreement with the results of *cis*-regulatory element analysis (Table S7). *GbNAC164* was up-regulated by MeJA and SA treatment. Moreover, *GbNAC164* keeps high expression level more durable by SA than by MeJA (Fig. 8). This result was consistent with *GhATAF1* (*He et al., 2016*). Thus, these findings suggest that up-regulating of the 62 *GbNAC* genes may result in increased or decreased VW resistance in cotton and these genes can be candidate genes for in-depth study on VW resistance.

Interestingly, *GhATAF1* was also up-regulated by ABA, cold, and salt treatments. Overexpression of *GhATAF1* confers transgenic cotton improved salt tolerance (*He et al.,*

*2016*). Similarly, Transgenic *GbNAC1* results in enhanced drought tolerance in Arabidopsis (*Wang et al., 2016*). The *NAC* genes *ANAC019*, *ANAC055* and *ANAC072* from Arabidopsis were up-regulated by drought, salt, and ABA treatments. Transgenic either *ANAC019*, *ANAC055* or *ANAC072* conferred Arabidopsis plants enhanced drought tolerance. The finding suggests that the 62 *GbNAC* genes may play role in response to abiotic stresses.

## CONCLUSIONS

In this study, the plant-specific *NAC* gene family in sea-island cotton was characterized with particular focus on their responses to VW infection. A total of 270 *GbNAC* genes were identified and characterized in sea-island cotton. The gene structure, chromosomal distribution, gene duplication, conserved motif, *cis*-elements, and expression profiles of the *GbNAC* genes were analyzed. Furthermore, expression profile analyses revealed that 62 *GbNAC* genes may play crucial role in response to VW infection. However, further functional data are required to evaluate each *GbNAC* gene. Overall, our results will provide new insights for plant engineering programs so that economically important traits for cotton can be developed, including improved resistance to VW.

### Funding

This work was supported by the National Natural Science Foundation of China (No. 31571755) and the Agricultural Scientific and Technological Innovation Project of Shandong Academy of Agricultural Sciences (CXGC2018E06 and CXGC2018B01). The funders had no role in study design, data collection and analysis, decision to publish, or preparation of the manuscript.

### Grant Disclosures

The following grant information was disclosed by the authors:
National Natural Science Foundation of China: 31571755.
Agricultural Scientific and Technological Innovation Project of Shandong Academy of Agricultural Sciences: CXGC2018E06, CXGC2018B01.

### Competing Interests

The authors declare there are no competing interests.

### Author Contributions

- Zhanji Liu conceived and designed the experiments, performed the experiments, analyzed the data, prepared figures and/or tables, authored or reviewed drafts of the paper, approved the final draft.
- Mingchuan Fu and Hao Li performed the experiments, analyzed the data, prepared figures and/or tables, approved the final draft.
- Yizhen Chen performed the experiments, authored or reviewed drafts of the paper, approved the final draft.

- Liguo Wang and Renzhong Liu analyzed the data, contributed reagents/materials/analysis tools, prepared figures and/or tables, approved the final draft.

## Data Availability

The raw data are available in the Supplemental Files.

## Supplemental Information

Supplemental information for this article can be found online at http://dx.doi.org/10.7717/peerj.7995#supplemental-information.

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
