# Peer review of "Systematic analysis of NAC transcription factors in Gossypium barbadense uncovers their roles in response to Verticillium wilt"

_PeerJ, doi:10.7717/peerj.7995_

## Round 0.1 · original submission · Major Revisions

Dear authors all reviewers and myself consider your work interesting and see merit on it. However, there are several major and minor points that need to be satisfactorily addressed. Also, there are large parts of the text have been copied from existing sources and I specifically recommend you to rephrase those paragraphs. It is also very important that you refer how to qPCR assays follow/ the MIQE guidelines.

Reviewer 1 ·

Basic reporting

no comment

Experimental design

no comment

Validity of the findings

no comment

Additional comments

1. Lines 84-86 in the introduction section to the manuscript mentioned that ‘the availability of diploid and tetraploid cotton genomes has allowed scientists to identify the NAC gene family members at genome-wide scale, such as 145 genes in G. raimondii (Shang et al., 2013), 141 in G. arboreum (Shang et al., 2016; Fan et al., 2018), and 283 in G. hirsutum (Sun et al., 2018). However, systematic analysis of NAC genes in G. babardense is still lacking’. Please explain the difference among G. raimondii, G. arboreum, G. hirsutum and G. babardense. In addition, the basic gene structure and chromosome location of NAC genes in G. babardense were performed in the manuscript. However, the functional evolution analysis of G. babardense is rare. The evolutionary relationship between G. babardense and NAC genes in G. raimondii, G. arboretum, G. hirsutum or other plants should be described in the manuscript, including phylogenetic tree, gene motif analysis and collinearity analysis. This section can be improved by referring to Liu et al. (Genome-wide analysis of the NAC transcription factor family in Tartary buckwheat (Fagopyrum tataricum)). BMC Genomics (2019) 20:113.
2. In the results section (line 290 to line 291, page 16), mentioned that most duplicated gene pairs (80.77%) demonstrated similar expression patterns. Please explain whether there is functional redundancy in this case.
3. In the results section (line 311 to line 312, page 17), why choose these 15 differential genes for qRT-PCR analysis. What is the basis of selection?
4. In the discussion section (line 321 to line 323, page 17), please explain the reason for the difference in the number of NAC genes in G. barbadense, G. hirsutum and G. raimondii.
5. The reference format needs to be unified, the page number information should be complete.
6. For real-time PCR experiment, please provide the information of PCR efficiency calibration curves for each pair of primers according to the MIQE guidelines (The MIQE guidelines: minimum information for publication of quantitative real-time PCR experiments.2009, Clinical Chemistry 55:4).
7. Why do the qRT-PCR experiments use actin as internal reference gene?

Reviewer 2 ·

Basic reporting

On the whole, the English writing of this manuscript is very clear, the expression is accurate, the experimental design is reasonable, the results are analyzed in place, the discussions are logical, the conclusion are also persuasive. The data and figures provided are clear and comprehensive, and the paragraph layout is also very organized. In any case, there are several minor errors and deficiencies in the manuscript that need to be supplemented and improved.

Experimental design

1. Materials and Methods (In line 147): the VW inoculation method should be briefly described, such as the amount of inoculum.
2. Materials and Methods (In line 154), the author chose the 18S rRNA gene as an internal control gene in qPCR. In fact, there are several genes, such as β-actin,GAPDH,18s-rRNA, can be selected. The relevant literatures should be cited to explain why 18S rRNA was chosen as the internal control gene.

Validity of the findings

3. Result: Line 202 should be moved to the discussion section, since that this sentence is the comparison between your founding and previous literature.
4. In the abstract (24 lines), the author believed that the GbNAC gene family had undergone strong purifying selection during the evolution. Thus, in the discussion section, this opinion should be discussed in more detail.
5. Other minor errors have been underlined on the manuscript.

Additional comments

This work had indeed filled the gap in the research on the NAC family genes of sea-island cotton. In particular, the work discovered 62 NAC genes that may be related to the disease resistance of sea- island cotton, which has important value for future cotton resistance breeding.

Annotated reviews are not available for download in order to protect the identity of reviewers who chose to remain anonymous.

Reviewer 3 ·

Basic reporting

The paper described a comprehensive genome-wide analysis of NAC genes from cotton. The experimental was well carried and the manuscript contains novel founding. However, I have some concern which the authors should revise
1- Please check the English all over the manuscript
ie: line contained the largest number should say the highest
line 284 Three types, do you mean three clusters or three patterns
line 343- Arabidopsis ANC ?? do you mean NAC.

Experimental design

2- Authors used RNA-seq data from G. babardense cv. 7124 they should give information about this cultivars if it’s sensitive or tolerant..etc
3-line 99 in the title: sea-island cotton genome
4- Why the number of motifs searched by MEME is 20. Since the study is informative and no further domain functional analysis they could limit only to 10 motif search
5- Why not using PLACE software for Cis-acting element search. Generally, PLACE and PlantCARE are both used.
6-line 146 from to line 149: not necessary details line147. Starting from where the plants until line 149 RNA-seq
7- In candidate gene expression analysis I have several concerns
- Why using roots sample and not in both leaves and roots.
- Why focusing on hormone treatment and not Abiotic stresses such as salinity, drought..etc
- Authors should use two resistance contrasting cultivars for RT-qPCR analysis
- Authors should justify the selection of their 15 candidate gene for RT-qPCR (Arabidopsis orthologs, RNA-seq..etc)
- Statistic analysis
8: identification of membrane-bound TF is very important because NTL NAC genes play an important role in stress response.
9- why using only NAC protein from Arabidopsis in the tree how authors defined the phylogenetic groups. Plant NAC are usually divided into groups a–h. please see Shen et al. (2009): BioEnergy Research 2, 217–232
10- Authors should only provide Cis-regulatory element for their candidate genes only and not for all NAC genes (Figure 7). Besides, the correlation between orthologous relationship, phylogeny, Cis element, and expression analysis should be further discussed.
11. Figure 8 is not necessary and should be removed

Validity of the findings

The publication contains novel information including gene discovery and characterization.

---

## Round 0.2 · Minor Revisions

Dear authors one of the reviewers need some clarification in your response to the comments provided. Please try to clearly specify your modifications made in the text

Reviewer 1 ·

Basic reporting

The English writing of this manuscript is very clear

Experimental design

The experimental design is reasonable

Validity of the findings

no comment

Additional comments

In the responses to questions 1, 2 and 6, I did not find the corresponding answer according to the number of lines you provided. Please further check the location information provided.

Reviewer 3 ·

Basic reporting

The revised manuscript is better than the first version and at this state is suitable for publication

Experimental design

All reviewers suugestion were considered which made this section more clear andd provided lacked information in the first version

Validity of the findings

The findings is very interresting and reported novel information about NAC family in cotton in response to stress

Additional comments

The manuscript could be accepted without further revision

---

## Round 0.3 · Minor Revisions

Thanks for carefully explaining all modifications made. After reading the manuscript you need to make additional changes.

1) The paper still needs editing for proper English. I recommend you contact an English editing service.

Examples from abstract that needs editiong: "As one of the largest and plant-specific gene families" --> "As one of the largest plant-specific gene families"; "almost all of them were suffered a strong purifying selection during evolution" --> "almost all of them exhibited strong purifying selection during evolution"; "cis-actiong " --> "cis-acting".
There are more throughout the text, please edit-proof it correctly.

2) The version of the genome downloaded needs to be specified.

3) The Conclusions are overstated. For example, "15 GbNAC genes tested using quantitative real-time PCR (qPCR) were found to be involved in methyl jasmonate (MeJA) and salicylic acid (SA) pathways". Differential is suggestive of involvement but does not prove it. Restate as the result ("..found to be up or downregulated in response to...") or "...suggested that they could be involved in...") . Also the analysis was not "comprehensive" as stated in the abstract (really no analysis could be comprehensive).

---

## Round 0.4 · accepted · Accept

I think that the English editing has improved the clarity of your manuscript